# *Virtual*^*CPR*^: Virtual Reality Mobile Application for Training in Cardiopulmonary Resuscitation Techniques

**DOI:** 10.3390/s21072504

**Published:** 2021-04-03

**Authors:** Francisco Javier García Fierros, Jesús Jaime Moreno Escobar, Gabriel Sepúlveda Cervantes, Oswaldo Morales Matamoros, Ricardo Tejeida Padilla

**Affiliations:** 1Escuela Superior de Ingeniería Mecánica y Eléctrica, Instituto Politécnico Nacional, Ciudad de México 07340, Mexico; fgarciaf1300@alumno.ipn.mx (F.J.G.F.); omoralesm@ipn.mx (O.M.M.); 2Centro de Innovación y Desarrollo Tecnológico en Cómputo, Instituto Politécnico Nacional, Ciudad de México 07700, Mexico; gsepulvedac@ipn.mx; 3Escuela Superior de Turismo, Instituto Politécnico Nacional, Ciudad de México 07630, Mexico; rtejeidap@ipn.mx

**Keywords:** cardiopulmonary resuscitation, virtual reality, mobile application, head mounted display, immersive system

## Abstract

Deaths due to heart diseases are a leading cause of death in Mexico. Cardiovascular diseases are considered a public health problem because they produce cardiorespiratory arrests. During an arrest, cardiac and/or respiratory activity stops. A cardiorespiratory arrest is rapidly fatal without a quick and efficient intervention. As a response to this problem, the *Virtual*${}^{CPR}$ system was designed in the present work. *Virtual*${}^{CPR}$ is a mobile virtual reality application to support learning and practicing of basic techniques of cardiopulmonary resuscitation (CPR) for experts or non-experts in CPR. *Virtual*${}^{CPR}$ implements an interactive virtual scenario with the user, which is visible by means of employment of virtual reality lenses. User’s interactions, with our proposal, are by a portable force sensor for integration with training mannequins, whose development is based on an application for the *Android* platform. Furthermore, this proposal integrates medical knowledge in first aid, related to the basic CPR for adults using only the hands, as well as technological knowledge, related to development of simulations on a mobile virtual reality platform by three main processes: (i) force measurement and conversion, (ii) data transmission and (iii) simulation of a virtual scenario. An experiment by means of a multifactorial analysis of variance was designed considering four factors for a CPR session: (i) previous training in CPR, (ii) frequency of compressions, (iii) presence of auditory suggestions and (iv) presence of color indicator. Our findings point out that the more previous training in CPR a user of the *Virtual*${}^{CPR}$ system has, the greater the percentage of correct compressions obtained from a virtual CPR session. Setting the rate to 100 or 150 compressions per minute, turning on or off the auditory suggestions and turning the color indicator on or off during the session have no significant effect on the results obtained by the user.

## 1. Introduction

According to data from the American Heart Association (AHA), more than 135 million deaths due to cardiovascular diseases are registered in the world each year [1]. In Mexico, according to the Instituto Nacional de Estadistica y Geografía (National Institute of Statistics and Geography, INEGI in Spanish, Figure 1) [2], a group of heart disorders caused 12.3 deaths per 10,000 inhabitants in 2019, Figure 1a.

In addition, an increase in the rate by 3.1 deaths from 2011 to 2019 is shown. This growing problem has been measured with a constant rate of increase since 2012, becoming leading cause of deaths from heart disease.

From Figure 1b, heart disease for 2019 accounted for more deaths in Mexico than cancer and homicides combined. Therefore, cardiovascular diseases should be considered as a health problem for the public interest. Cardiorespiratory arrest is a phenomenon that occurs as a consequence of these diseases; during an arrest, cardiac and/or respiratory activity stops. It is a situation where resuscitating the victim is possible when intervening quickly and effectively. However, it is known that 80% of cardiac arrest situations occur at the victim’s home, hence, far from hospital and expert care, the death rate is greater than 90% [3].

This is where the importance of first aid lies, which is the set of theoretical–practical knowledge for the care of a health emergency while waiting for professional help. Part of this knowledge is the cardiopulmonary resuscitation technique (CPR), a series of maneuvers that, learned by the general population and applied in a timely manner, will increase the chances of keeping the victim of cardiac arrest alive [3]. Starting from the need to acquire basic knowledge in first aid, training is used as a tool, made up of didactic activities for learning and development, being the means for the acquisition of these skills.

Quality of CPR is considered an avoidable damage in the care of cardiorespiratory arrests, so a standardized approach is used to improve results. Part of the approach is the effective teaching of the basic technique to the general public. In the non-experts, training is necessary to assist in these extraordinary cases as the appropriate dynamics and materials must be used [3]. Part of the limitations for the personnel training in CPR techniques are those related to the instructors availability and specialized material. Although the technique is standardized, it requires effective theoretical–practical teaching for its correct application in a real case.

Virtual reality is a group of technologies that allows to simulate experiences superior to the real world. Due to its ability to see, touch and hear artificial environments, virtual reality has an important market segment of 18.8 billion dollars worldwide in 2020, according to data from Stadista in [4]. Fields of application for virtual reality include education; for instance, review and analyze methodologies for Science, Technology, Engineering and Math (STEM) learning is analyzed in [5], highlighting how the use of virtual reality can play an important role in the learning process. Alizadehsalehi et al. in [6] developed an application within the Architecture, Engineering and Construction (AEC) industry—highlighting the role of extended reality, a variant of virtual reality, to simulate construction projects and using digital models to visualize the phases of the process. Virtual reality also represents a development field for applications, including medical ones, e.g., described the evolution of rehabilitation in physiotherapy by presenting exercises interactively, increasing the motivation of the patient and reducing the work of the physiotherapist. Lee and Lee in [7] suggest that virtual reality can be used to create an exercise program for patients with spinal cord injury.

This project is defined as *Virtual*${}^{CPR}$, and it has been developed to measure the CPR performance on training mannequins by using a sensor that records the strength of the compressions applied by the user during the session. *Virtual*${}^{CPR}$ is a small, accessible and portable device that allows communication with a smartphone, and it is intended by its manufacturers to develop an application for this platform. Furthermore, a virtual reality mobile application is an accessible, portable and interactive tool that offers complementary tools to CPR training sessions. Virtual reality allows immersive virtual experiences to be deployed within a virtual world developed and designed according to the needs of the training process.

In addition, this work is related to the training of the standardized technique of cardiopulmonary resuscitation. There are two types of cardiopulmonary resuscitation, namely basic and advanced. The former is the indispensable knowledge about the technique that will be used to carry out the project, and the latter is the use of specialized resuscitation instruments, which is discarded by the high-requirements for its teaching. Starting from the basic technique, the project focuses on the steps for an adult resuscitation using only the hands because it is the minimum training required to act in case of witnessing a cardiac arrest.

For the development of this proposal, we work with virtual reality technology due to it being a whole system with the ability to simulate the interactions between a user and a virtual world. Moreover, there is augmented reality that seeks to supplement the real world with virtual objects without necessarily creating an immersive environment. Both technologies are similar since they seek that the user perceives and interacts with the virtual elements, but different because of the hardware resources and techniques used to implement them. The development of a virtual reality solution is chosen over augmented or mixed reality, the latter a combination of both concepts, due to the technological challenge of implementing them on mobile devices targeted by this project.

For user input interactions, touch screen used and body movements are features chosen over others such as voice, mouse and keyboard commands, due to to CPR technique requirements. The selection of a native application for the Android operating system is based on the characteristics for its development and publication, since other operating systems for mobile devices require additional conditions for their use. The use of a web application is ruled out to overcome its limitations in access to the device’s hardware, necessary to work with virtual reality technology.

The hypothesis of this work is that a virtual reality mobile application is necessary to help learning and practice of the CPR technique for users with and without previous training. Therefore, this proposal implements an interactive virtual scenario which would improve the CPR technique with continuous use of the *Virtual*${}^{CPR}$ system. Nonetheless, the mobile application of this type would not achieve its objective of helping the user to learn and improve their CPR technique if continuous practice and evaluation are omitted, which depends directly on the user’s motivation to dedicate time to use the application again, so incentives are required in the design of the training that present reasons for using them more than once.

*Virtual*${}^{CPR}$ is based on the cardiopulmonary resuscitation technique, an intervention procedure that allows a person to be kept alive when they are in cardiac arrest. Basic technique training commonly requires a trainer, a designated space and the use of mannequins for practice. In addition, there is virtual reality, which is a set of technologies that are growing as an alternative for training in first aid techniques such as CPR. Hence, there is a mix of user interactions in the real world with a virtual world designed for teaching, using scenarios and virtual resources to simulate real situations. In addition, mobile devices such as tablets and smartphones use applications designed to facilitate a specific job; thus, the software resulting from this project will be designed as a mobile application compatible with these devices.

This paper is divided into six sections. Section 2 describes and compares the characteristics of the related work, which will lead to a discussion on the differentiator of the proposed solution. In Section 3, the medical and technological knowledge required in the development of training in the CPR technique are integrated. In Section 4, the procedures for the development of the *Virtual*${}^{CPR}$ are described in detail. In Section 4, *Virtual*${}^{CPR}$ methodology is described in detail. Experimental tests are carried out in Section 5 for verifying the correct operation of the system. Finally, the conclusions obtained are discussed in last section.

Our findings point out that the more previous training in CPR a user of the *Virtual*${}^{CPR}$ system has, the greater the percentage of correct compressions obtained from a virtual CPR session. Setting the rate to 100 or 150 compressions per minute, turning on or off the auditory suggestions and turning the color indicator on or off during the session have no significant effect on the results obtained by the user.

## 2. Related Work

In order to compare the present work with the state-of-the-art works, databases from sites such as Hindawi, IEEE Xplore or the Multidisciplinary Digital Publishing Institute (MDPI) were used. Regarding the topic of cardiopulmonary resuscitation, the works of Vaughan et al. in [8], Santos et al. in [9], Durai et al. in [10], Habibian et al. in [11], Dumcke et al. in [12] and Stroop et al. in [13] are technological solutions when teaching and practicing CPR technique. The most important works found on the virtual reality topic were defined by Vaughan et al. in [14], Durai et al. in [15], Liyanage et al. in [16], Everson et al. in [17] and Semeraro et al. in [18]. These five works are medical applications for this branch of technology. This work makes use of VR mobile applications for training of CPR. Therefore, *Virtual*${}^{CPR}$ can be related or compared with the works of Leary et al. in [19] and Nas et al. in [20]. However, the works proposed by Vaughan et al. in [8], Vaughan et al. in [14], Durai et al. in [15], Liyanage et al. in [16], Everson et al. in [17], Leary et al. in [19], Semeraro et al. in [18] and Nas et al. in [20] are closer related to our proposal since all of them are both a solution for teaching CPR and systems based on virtual reality.

Everson et al. in [17] carried out a literature review, highlighting virtual reality (VR) and augmented reality (AR) technologies to simulate or improve CPR training environments. This work presents the haptic as the tactile element of virtual experiences that provides feedback to force interactions, concluding that the challenge of immersive training systems is to compensate for their limitations of mechanical complexity and cost. Leary et al. in [19] studied the implications of mobile virtual reality by comparing a CPR virtual reality application with a conventional CPR technique. The authors compare the user’s response regarding the necessary steps in CPR and the depth of compressions, concluding that there is an improvement when following instructions. Semeraro et al. in [18] compared a virtual reality application with the use of mannequins to teach CPR, as a result they found equivalence in the performance of both strategies for training programs that implement gamification in their design. Finally, Nas et al. in [20] designed a test to compare the performance of CPR learned with a virtual reality application against face to face training, concluding that it is important to test virtual reality alternatives to achieve results similar to traditional training.


Vaughan et al. in [8] developed a virtual reality simulator for training in the CPR procedure with children in a school environment by using *Oculus Rift* hardware and the *Leap Motion* sensor to implement the simulator used for teaching in schools with children, in order to place the system user in a virtual situation constructed and guiding on the correct application of the resuscitation technique. Moreover, this proposal uses a mannequin to complement the simulator, 3D modeling of environments and objects to create the virtual world, and the use of *Leap Motion* keeps the hands free while recording the user’s movements so that their actions take place in the simulation. The authors maintain relationships with organizations such as the British Heart Foundation, Cymru Wales Ambulance Service, NHS Trust and the College of Paramedics (UK), in order to give expert feedback to their system and obtain clinical evaluation. This CPR virtual reality training simulator for schools is a system that requires the user’s movements as input data through the *Leap Motion* sensor assembled on the virtual reality viewer, as well as the angle of view and the position. The simulator takes the information and updates the virtual world with the interactions carried out to deliver a stereoscopic image of the virtual environment. Finally, the process is repeated to keep virtual reality in constant change according to the motor instructions of the surgeon.

Vaughan et al. in [14] developed a virtual reality prototype for paramedic simulation training, using state-of-the-art hardware for the practice of complex procedures within the simulation. This proposal, based on its portable *Oculus Quest* system, explores the feedback force characteristic to create the realistic feeling of being punctured during procedures, and simulates advanced procedures such as Needle Cricothyroidotomy and chest decompression procedure. The product is a functional prototype for the *Oculus Rift* viewer that allows interaction with the simulation of medical procedures, and it is based on the manipulation of a virtual human model of the patient in three dimensions to simulate the correct location of instruments on several nodules. The virtual reality system records the viewing angle along with the position and commands of a manual controller as input data to generate an image in the virtual reality tool. The image is finally displayed on the *Oculus Rift* glasses and repeats the process to keep the user in constant interaction with information and virtual objects to simulate procedures.

Durai et al. in [15] generated a virtual reality system for the simulation of the CPR procedure. The sensors for tracking position and the optical device to display the image are part of the *Oculus Rift* hardware system, requiring specialized computer equipment and a force detection plate and, therefore, the use of a mobile application or its deployment in scenarios lacking specialized hardware. Hence, it is an application designed for training. The methodology of this system starts with a force sensor recording the user’s compressions as the main interaction with the simulation, and the system provides information during the procedure using an interface that finally reaches the *Oculus Rift* device as a stereoscopic image of the virtual world.

Finally, Liyanage et al. in [16] developed a virtual reality simulator for training in the cardiopulmonary resuscitation procedure. The procedure requires the user to have a perception of the strength used during compressions to the victim’s body; the system includes force sensors to record this information. As this proposal relates to a medical application designed for specialized hardware, the use of a mobile application is not required either. This system is a cycle that begins with the measurement of the force applied to a sensor integrated in a training mannequin. The information of the user’s interaction goes into the system that updates a virtual model visualized by the user via the *HTC Vive* system. When a mannequin is modified, the *HTC Vive* virtual reality system and a computer are required for graphic rendering of the virtual stage.

Table 1 compares the features of the related works and the proposal presented here. These works were chosen for their relationship with the branch of medicine and the use of virtual reality technologies, therefore, we present some differences and similarities.

In order to implement the virtual world in each case, commercial virtual reality systems are used. The most used system in the work group is *Oculus*, a system that integrates glasses, controllers and sensors for the development of virtual reality experiences. Additionally, computer equipment capable of generating simulation graphics in real time is required.

This work is intended for the general public who want to learn and practice CPR. Hence, this proposal only makes use of three fundamental elements: (i) a smartphone with an Android operating system, (ii) a portable force sensor and (iii) mobile virtual reality glasses. Additionally, pre-rendering of the objects that makes up the virtual stage, a level of photorealism is reached close to that of the systems with the highest-cost and graphic power. These components could allow the *Virtual*${}^{CPR}$ application to be used by anyone, anywhere at a low-cost in order to simulate a virtual and immersive experience.


## 3. Materials and Methods

### 3.1. Cardiopulmonary Resuscitation

Cardiopulmonary resuscitation (CPR) technique standardized by the AHA is an intervention procedure that aims to keep a person alive while they are in cardiac arrest [1]. CPR is part of the first aid preparation, i.e., a set of theoretical and practical knowledge, and its intention is to guide in the assistance of a health emergency. The timely and quality application of the technique will keep a person alive whose heart and breathing have stopped functioning until they can receive adequate care.

Depending on its complexity, the standardized CPR can be of two types: (i) basic and (ii) advanced.

On the one hand, the basic technique has the following essential components:5 cm deep chest compression for adults and children, and 4 cm for breastfed babies.Frequency of 100 to 120 compressions per minute.12 rescue breaths per minute.Evaluation and call to the local emergency number.

On the other hand, the advanced technique requires specialized training for additional procedures such as:Adrenaline injection.Defibrillation.Recovery of vital signs.Diagnosis of terminal illness.Airway optimization.Neurological control.Metabolic control.

Likewise, the technique is applied differently according to three age groups: (i) adults (end of puberty and onwards), (ii) children (over one year and during puberty) and (iii) breastfed babies (under one year).

Another important factor is the environment under which the technique is performed, since it is related to the level of personnel and equipment available:Out-of-hospital, which is made up of any space outside a hospital, either the public thoroughfare, recreational spaces, workplaces or homes.Intra-hospital, a hospital that can provide medical care, surgical operations, and hospital stays [21].

Figure 2 shows a six-step methodology when applying the standardized cardiopulmonary resuscitation technique. In this way these steps are the following:Check that the scene is safe. You should look around to make sure that something could not injure you or the victim.Feel the victim’s shoulders while addressing his/her aloud, then check if the person responds and if there is no response continue.Shouting for help, if someone comes ask to call the emergency number, otherwise, call personally.Check his/her breathing and rhythm, to do this:Make sure the victim is lying on a firm and flat surface.Check his/her breathing.Check the heart rate.If the victim has no pulse and is not breathing or is only panting, perform CPR.Perform compressions and rescue breaths.To keep the victim alive you should continue the series (30 compressions + 2 rescue breaths) until the victim begins to breathe or move, or until someone more trained arrives and takes over [21].

The AHA’s recommendation regarding the intervals to check the heart rhythm is two minutes. This indicates that the compressions should be continuous for the recommended time interval and that we should pause quickly to check the vital signs [22].

In addition, AHA’s recommendation regarding the depth of the compressions, at the spot that is vertically over the sternum, is 5 cm. The force necessary to reach the level depends on the characteristics and weight of the victim, on average a force of 500 Newtons is equivalent to 50.98 kg [23].

### 3.2. Virtual Reality

Definitions in the field of virtual reality are constantly changing as it is a recent technology, the following two definitions are the most common:Virtual reality consists of systems that use computer-generated scenarios to simulate interactions between the user and a virtual environment in real time, defined by Forcael et al. in [24].Virtual reality is the technology that provides an almost real and credible experience in a synthetic or virtual way. In practice, virtual reality is the combination of specialized hardware and software. This technology is continuously in development and although it is normally associated with entertainment, it can be applied as a diagnostic or therapy tool, defined by Rutkowski in [25].

The previous two definitions mention systems with the ability to generate views of a virtual world, using computer generated graphics and specialized hardware consisting of controllers, position sensors and stereoscopic helmets. It is possible to define virtual reality without talking about hardware as long as the concepts of presence and telepresence are considered [26].

Presence can be understood as physically experiencing an environment, it refers to one’s own perception of the surroundings. Presence is the feeling of being in an environment. There are many perceptual factors involved in achieving the sense of presence, limited by the space that is perceived, but also by human sensory organs.

In this way, the term of telepresence can be used to describe the mediated sensation of a second environment over a first one, immediate and physical environment. Sherman and Craig in [27] define that: Telepresence is defined as the experience of presence in an environment through a means of communication. The difference between the concept of presence and telepresence lies in the origin of the perceived environment, presence refers to a natural and immediate perception, while telepresence refers to the perception mediated by a communication channel. The communicated environment can be temporally or spatially remote and can be a captured real environment or a virtual creation. In seeking to experience virtual environments as is done with the physical world, the same perception mechanisms must be applied. Liveliness and interactivity are the most determining factors in achieving telepresence, they represent a technological challenge and in conjunction with the perceiver.

From Figure 3, *Liveliness* is a property of technological means that has an influence on the induction of the sensation of presence. It means the richness of the representation of an environment defined by its formal characteristics, i.e., the way in which information is presented to the senses, consisting of two variables:Amplitude refers to the number of simultaneous sensory dimensions that are presented.Depth refers to the resolution of the perceptual channels, usually associated with quality.

In addition, *Interactivity* is the participation of a user in the modification of the form and content in an environment when the user perceives it in real time. It is the combination of communication variables of the user with a perceived environment. Interactivity is made up of three main variables:Speed refers to the rate at which an input variable can be assimilated.Range refers to the number of possible actions in a given time.Mapping is the ability of a system to control changes in the environment in a natural and predictable way [26].

## 4. Design of *Virtual*${}^{CPR}$

Figure 4 shows the *Virtual*${}^{CPR}$ system, which is a mobile application whose inputs are the force applied by the user to perform the compressions of the CPR technique and interactions through buttons and physical movements during the simulation. The system delivers a stereoscopic image at its output, as an updated window to the virtual world that responds to user actions. The system has a feedback block where the user obtains information from the simulation and modifies the inputs in response.

Internally, the *Virtual*${}^{CPR}$ system uses the *HX711* integrated circuit block, an analog–digital converter that, together with the sen-1045 sensor, measures the force applied to a mannequin or substitute. At the output, it delivers a constant flow of measurements by serial communication with the *NODEMCU ESP32* microcontroller. A program is designed for the microcontroller that receives the serial data from the converter and, using its transfer function, obtains the equivalence in kilograms. The microcontroller is configured as a *Soft Access Point* for a network name, accessible only wirelessly and without passwords for access. This configured server detects the connection of a client and sends the information through WiFi to the device that generates the stereoscopic image.

Equation (Equation 1) is the transfer function of the *ESP32* microcontroller *HX711.h* library for data conversion. Therefore, a polynomial adjustment is necessary with values obtained experimentally from the output of the *HX711* converter. First, three series of measurements were averaged for known weights between 0 and 40 Kg, then the data were plotted to observe the behavior of the sensor. Finally, the linear trend of the graph leads to proposing a valid first degree polynomial from 4.8165 Kg.
(1)$$y=\frac{x}{-32103}+4.8165$$

In addition, Figure 4 depicts the sensor block and analog–digital converter. In this way, we employ *Sen-1045*, which is a load cell sensor for being configured as half a *Wheatstone* bridge, and the *HX711* converter for being a 24-bit ADC converter with serial data output. The set described is used as part of a circuit for the measurement of the force variable. Hence, the bridge is completed using 1 kΩ fixed resistors and the millivolts generated by the deformation of the sensor are the input of the analog–digital converter. Thus, the resulting binary data are sent by serial $I^{2}C$ communication to the smartphone system block.

Smartphone is the device assigned to receive the force data, as well as the software and hardware platform necessary to make the simulation work. It requires user interactions in the form of movements and the indirect touch of the screen, which are read by the integrated sensors such as the gyroscope, the accelerometer and the touch screen characteristic of each model and manufacturer. Scripts are used for communication between the microcontroller and the smartphone, which are C# language programs for the *Unity* simulation and game creation engine. The script creates a thread to receive the force values upon input. The configuration of a thread allows the processor to take advantage of its resources to keep network communication, while the simulation is developed separately thanks to the *Unity* engine.

The development of a simulation for CPR training using virtual reality techniques requires interaction variables with the user in the physical world. The input variables are the force values in units of mass, the user’s interactions variables and the movements of the user during the session, Figure 5. To reflect the user’s actions, a final conversion stage is required for use in modifying the simulation. The main interactive elements are four and they can be defined as follows:
First, the user’s interface is made up of panels, images, sounds and buttons that help out the user to orientate himself/herself to control the simulation by using a series of conditions for the variables that determine whether or not an action is taken, Figure 5a.Secondly, the model is a mannequin in three dimensions depicted by Figure 5b, and it is built using the free-human modeling software Make Human Community. The purpose of the mannequin is to simulate an individual who requires CPR, responding to the compression values applied by the user and helping to locate the actions relative to its position.As output from second interface, we present a third interface (Figure 5c), consisting of user instructions for application, mainly display of menus, session options and configuration of virtual experience. Interactions between force sensor and mannequin generate animations visualized by the user, helping him/her to perceive his/her actions within the simulation.To complete the experience of a virtual world, an artificial stage is rendered to recreate a room of our University, Figure 5d. The result is a 360-degree static image surrounding CPR scene and configuring lighting and dimensions to simulate a space during training. For the modeling process, Blender (a free and open source software) is used as a tool for animation, visual effects and 3D modeling.

## 5. Experimental Results

### 5.1. Experimental Setup

Recommended hardware features to use the *Virtual*${}^{CPR}$ mobile app are:Smartphone with an Android 4.4 or higher operating system.Screen resolution: 1920 × 1080 pixels.CPU speed: 1.4 GHz or higher.RAM: 2 GB or higher.Integrated gyroscope sensor.Integrated accelerometer sensor.Virtual reality viewer according to the physical dimensions of the smartphone.CPR training mannequin.

### 5.2. Performance Test

A performance test is proposed to demonstrate the use of the features available in the *Virtual*${}^{CPR}$ mobile application, Figure 6. For carrying out these tests, a mannequin for CPR practice is integrated with the *Sen-1045* force sensor, see Figure 6a. This is located on the pressure plate in the center of the mannequin’s chest as shown in Figure 6b. Subsequently, the upper layer of the mannequin is placed as shown in Figure 6c.

In this way, this test to verify the general functioning of the *Virtual*${}^{CPR}$ system is defined in the following four steps, Figure 7:*Measurement of input variables* verifies the correct operation of the force sensor when its values are reflected in the simulation by means of indicators and animations, Figure 7a.*Wireless communication* is performed by means of a force sensor along with a microcontroller, a prior connection to the *RCP_Simulator* network is required, Figure 7b.*User’s interface* is the set of control elements, menus and indicators during a CPR session to verify its correct performance. In addition, it is necessary to carry out a complete *Virtual*${}^{CPR}$ session, Figure 7c.*Interactivity* allows an interaction between the *Virtual*${}^{CPR}$ system and the final user. In this way, this proposal was put to the test by completing the necessary steps for a complete CPR session, checking at all times the effect of compressions on the elements without delay perceptible by the human being (0.5 s), Figure 7d.

### 5.3. Experimental Design

Experimental design aims to determine what variables are significant and, therefore, what variables could help to learn and practice the CPR technique by calculating the percentage of correct compressions obtained in a *Virtual*${}^{CPR}$ session. Hence, we consider the following variables to be treated:Previous training in the basic technique of cardiopulmonary resuscitation.Frequency of compressions marked by the application.Listening suggestions during the session.Color indicator during the session.

The following experiment is based on an analysis of variance (ANOVA) to determine what variables have the greatest influence on the response. The selected sample is made up of two not-trained users in the basic technique of CPR and two trained users, practitioners and trainers in first aid techniques including CPR. The sessions lasted two minutes and are configured according to the combination of values for the four factors involved. Each factor can take its high or low value so the number of combinations is calculated as number of tests and is equal to $2^{Factors}$, Table 2. Therefore, the number of combinations for the experiment is 16 tests and 2 test blocks, so the total number of sessions to be analyzed is 32 complete cases.

### 5.4. Results

For this experiment, we employ a multifactorial ANOVA. This procedure runs a multi-factor analysis of variance for the percentage of correct compressions. It performs various tests and graphs to determine which factors have a statistically significant effect on the percentage of correct compressions. It also assesses the significance of the interactions between the factors, if there is enough data. The F-tests in 32 complete cases of ANOVA allow us to identify the significant factors. For each significant factor, the multiple range tests will tell users which means are significantly different from others.

On the one hand, the maximum percentage obtained during the experiment was 38%, belonging to the set of tests carried out by experts, with a frequency of 150 compressions per minute, using the auditory suggestions and deactivating the color indicator during the session. On the other hand, the lowest percentages recorded with a value of 0% are distributed in the tests for both groups of users and different combinations for the other variables. It is possible to highlight 6 out of the 8 lowest percentages were obtained by not-trained users, Figure 8.

Figure 8a shows that the category of trained users concentrates the highest percentages of correct compressions, pointing out the most significant factor according to multifactorial ANOVA.

In Figure 8b is depicted the significant or non-significant relationship for the experiment of the four-variables to consider. A short distance between the low and high values of the first three variables (indicator, suggestions and frequency) indicates a non-significant relationship, and the greater distance between the two values for the variable previous training points out greater significance on the percentage value of correct compressions.

F-Ratio is used to accept or reject a hypothesis. For an ANOVA table, the hypothesis $H_{0}$ is the statistical significance of a variable regarding to the others and $H_{1}$ is the non-significance; the higher the value of the reason F, the greater the significance of the variable. The *p*-value is the probability that a variable is statistically significant; the lower the *p*-value, the more significant a variable is. Thus, an arbitrary level of significance associated with a confidence percentage must be set.

Table 3 shows the variability-decomposition percentage of correct compressions into contributions due to the factors: previous training, frequency, suggestions and indicator. As the Type-III sum of squares has been chosen, the contribution of each factor is measured by removing the effects of the other factors. The *p*-values test the statistical significance of each of these factors. When a *p*-value is less than 0.05, the factor to be treated has a statistically significant effect on the percentage of correct compressions with a 95.0% confidence level. Therefore, the *p*-value of the previous training factor is less than 0.05, indicating 95%-statistical significance in the confidence interval, while the rest of the factors analyzed are considered non-significant. Hence, a second multifactorial variance analysis is performed, eliminating non-significant factors.

The elimination process must be carried out until all the significant factors are found. For the analysis of the previous training (frequency, suggestions and indicator), only the previous training factor proves to be significant. Table 4 shows the second analysis that eliminates the effects of the other variables, resulting in a *p*-value equal to 0.0006 (less than the arbitrary level 0.05) and with the level of the confidence interval at 95%, pointing out that the significant variable is over the percentage correct compressions. In summary, the more previous training in the basic technique of cardiopulmonary resuscitation a user of the *Virtual*${}^{CPR}$ system has, the greater the percentage of correct compressions obtained from a virtual CPR session. Setting the rate to 100 or 150 compressions per minute, turning on or off the auditory suggestions and turning the color indicator on or off during the session have no significant effect on the results obtained by the user.

As a result of experimenting with *Virtual*${}^{CPR}$, it was possible to observe the importance of the space where the CPR session takes place, i.e., it must be free of obstacles interfering with the perception of the virtual world and remain consistent with what the user sees and touches. Regarding the system-feedback against interactions with the user, during the tests carried out the tested-subjects expressed their need to constantly know their performance, supporting the existence of visual and auditory indicators as part of the user interface.


### 5.5. Implications and Limitations

A virtual reality mobile application is a tool with the advantages of an accessible cost, portability and interactivity with the user. *Virtual*${}^{CPR}$ offers complementary features to the CPR training sessions, creating a virtual world that can be adapted to the needs of the teaching process. The main challenges when implementing a virtual reality mobile application are the motivation of the user to acquire and use it. As these kinds of applications are for basic knowledge in first aid, they are essential for the general public, thus the attractive and recurrent design of training dynamics becomes important.

According to the above, the practical implications of *Virtual*${}^{CPR}$ are initially that the user is not familiar with virtual reality, it is a growing industry that has not yet reached all users, in the same way the dynamics of interaction with the system require continuous practices before coming to people’s full knowledge. A need to highlight is to have a clear and flat space to perform the technique correctly. It is also possible that the correct level of force is not reached for all compressions during a session, it is a process of physical adaptation to perfect an effective technique that keeps a real victim alive. Finally, quality adjustment is required on devices with different levels of hardware to maintain real-time performance.

Moreover, the present work was designed considering a set of limitations. The first one is a design based on virtual reality technology, without ruling out other related technologies in the future, such as augmented or mixed reality, that are functional and affordable. The second one, the development for Android devices, represents the appropriate platform to experience virtual reality, other operating systems require additional development to use *Virtual*${}^{CPR}$, and it is also a requirement to use a smartphone with Android 4.4 or higher to use virtual reality. Finally, the basic CPR technique using only the hands was selected as the initial training offered by this proposal, it is also necessary to add more initial and advanced techniques to enrich learning at all levels of training.

Lastly, based on the results of designing and presenting *Virtual*${}^{CPR}$, the following implications can be considered [28,29]:Study of the effects of virtual teaching of the CPR technique in sessions assisted by professionals or users with the interest of first aid self-learning.Professional training for users through virtual experiences modeled to approach simulated cases of risk in real life.Motivation of researchers to develop more and better mobile simulators that find opportunities in the benefits of recent mobile devices, designed to implement virtual reality.

Due to the implications of the pandemic caused by the SARS-COV2 virus, it is impossible going back to the laboratory facilities to increase the number of participants and then the number of experiments.


## 6. Conclusions

Seven out of the several works related to the virtual simulation of the cardiopulmonary resuscitation technique (CPR) were selected to perform our analysis because they belonged to a branch of applications based on virtual reality for teaching and training in CPR. These seven works were found with the main differences being the design approach to their simulations and the platform used to be developed.


Medical knowledge in first aid, related to CPR for adults, and technology, related to the development of simulations on a mobile virtual reality platform, were integrated to define the alternative in CPR training.

The development processes for the virtual reality mobile application were presented using the *Virtual*${}^{CPR}$ methodology, developed in three main processes: (i) obtaining the data with sensors and conversion to use them in the simulation, (ii) data transmission wirelessly from a microcontroller to a mobile device and (iii) simulation of a CPR scenario within the mobile device and its visualization using stereoscopic glasses.

To verify the functioning of the *Virtual*${}^{CPR}$ system, a functional test was performed and an experiment of 32 tests was designed. Four factors are considered for a CPR session: (i) previous training in CPR, (ii) frequency of compressions, (iii) presence of auditory suggestions and (iv) presence of the color indicator. The outcomes obtained were analyzed by the multifactorial analysis of variance, finding that the user’s previous training in CPR has a significant effect on the percentage of correct compressions obtained from a *Virtual*${}^{CPR}$ session.

Therefore, a virtual reality mobile application was developed in order to train in the technique by developing an application for a mobile platform. This proposal is based on the design of a virtual environment that instructs in the basic steps of the technique while deploying timely information to the user. In the present work, a virtual reality mobile application was designed for the general public, trained or not-trained user in practice of basic CPR. User’s interactions with the system are through a portable force sensor for integration with training mannequins, whose development is based on an application for the Android platform.

## Figures and Tables

**Figure 1 sensors-21-02504-f001:**
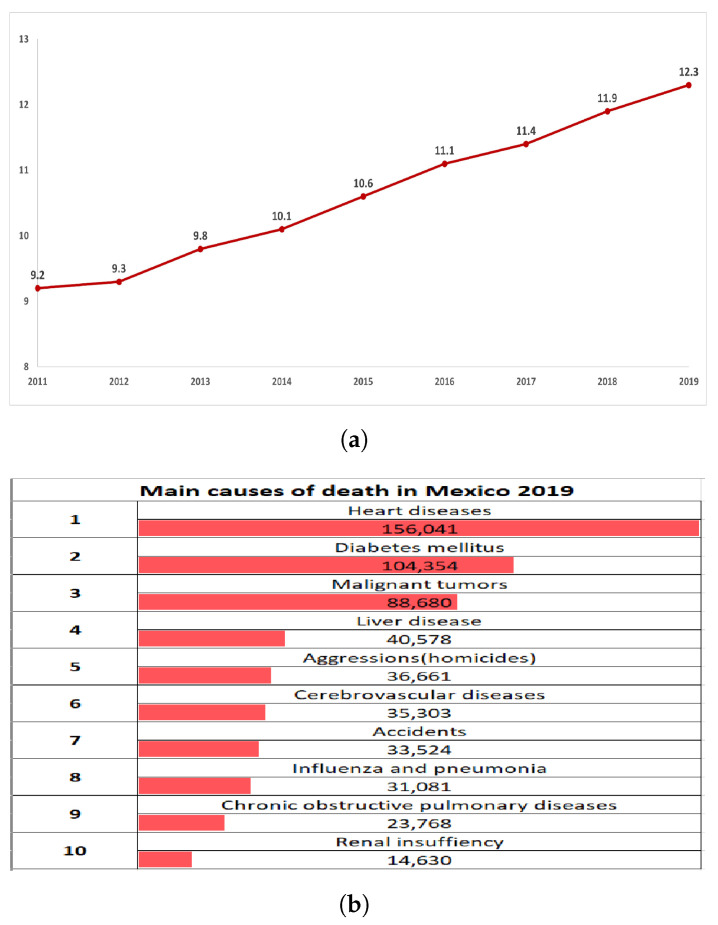
Data from the National Institute of Statistics and Geography on diseases in Mexico. (**a**) Death rate registered due to heart disease per 10,000 inhabitants from 2011 to 2019 and (**b**) main causes of death in Mexico for 2019.

**Figure 2 sensors-21-02504-f002:**
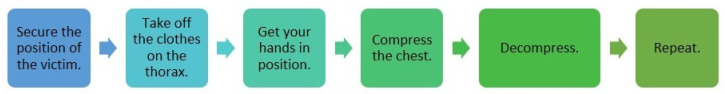
Application methodology for the cardiopulmonary resuscitation (CPR) technique.

**Figure 3 sensors-21-02504-f003:**
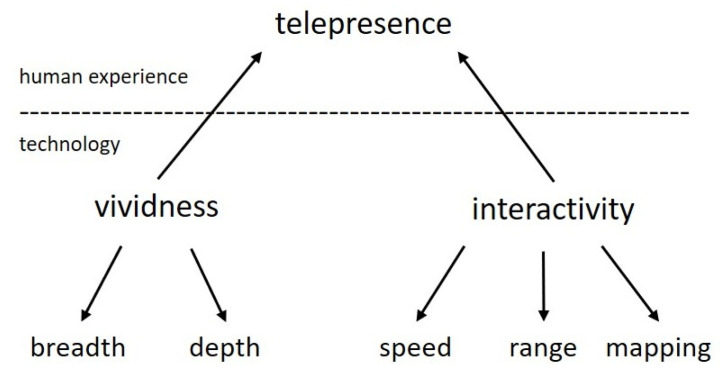
Technological variables of telepresence.

**Figure 4 sensors-21-02504-f004:**
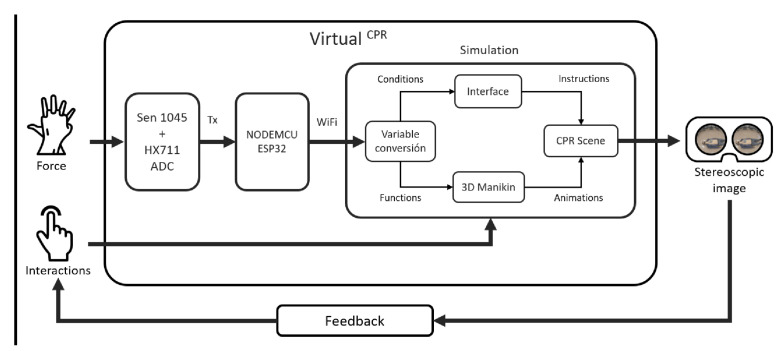
Internal blocks of the *Virtual*${}^{CPR}$ system.

**Figure 5 sensors-21-02504-f005:**
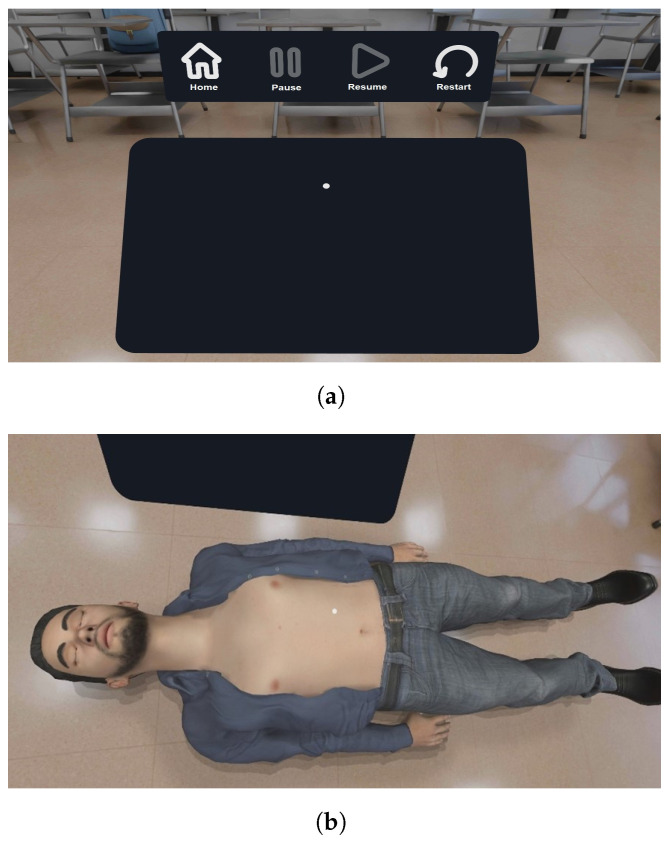
Main interactive elements of the *Virtual*${}^{CPR}$ system. (**a**) Interface during a session, (**b**) virtual mannequin model for the simulation of compressions, (**c**) instructions during session and (**d**) virtual environment modeled for the proposed system.

**Figure 6 sensors-21-02504-f006:**
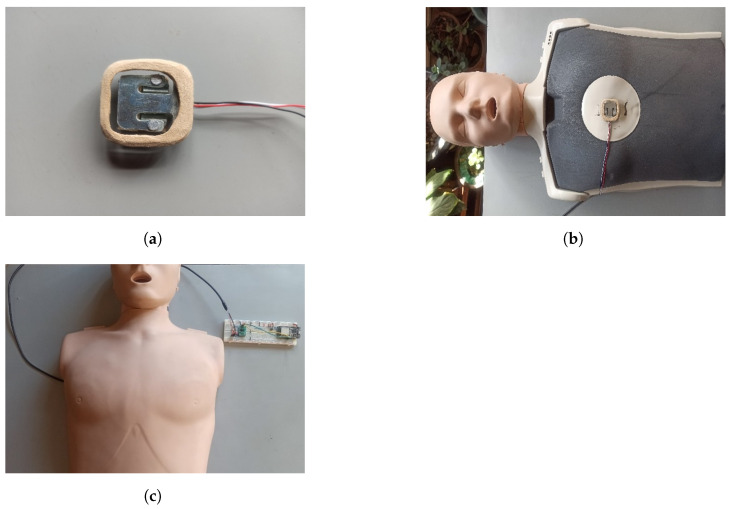
Integration of the *Sen-1045* force sensor into a mannequin for CPR. (**a**) *Sen-1045* force sensor, (**b**) position of the *Sen-1045* sensor with respect to the training mannequin and (**c**) integration of the *Sen-1045* sensor and the training mannequin.

**Figure 7 sensors-21-02504-f007:**
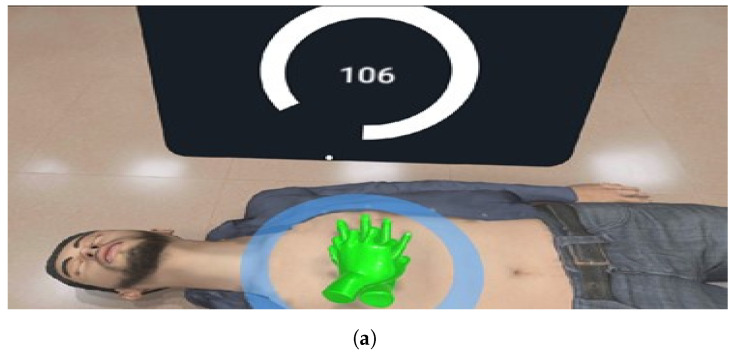
General functioning of the *Virtual*${}^{CPR}$ system. (**a**) Range operation of the force and effect sensor in the simulation, (**b**) wireless connection through the *RCP_Simulator* network, (**c**) result interface of a complete *Virtual*${}^{CPR}$ session and (**d**) test session with the mannequin.

**Figure 8 sensors-21-02504-f008:**
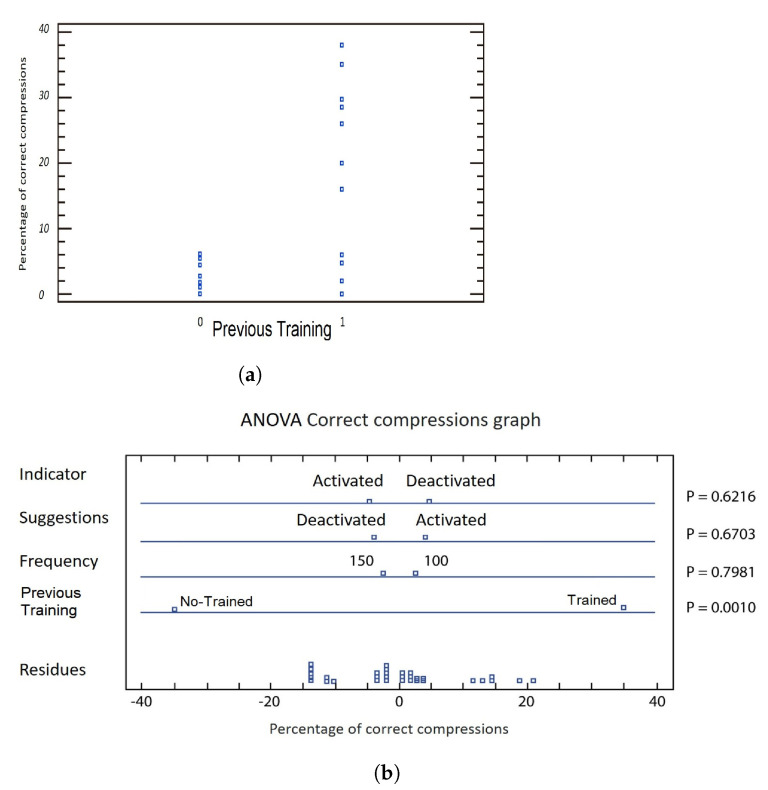
Main results. (**a**) Scatter plot by Level Code for the previous training variable, and (**b**) ANOVA percentage graph of correctcompressions.

**Table 1 sensors-21-02504-t001:** Main features of related works for comparison of related systems against this proposal.

Features	Vaughan et al. [8]	Vaughan et al. [14]	Durai et al. [15]	Liyanage et al. [16]	Leary et al. [19]	Semeraro et al. [18]	Nas et al. [20]	Proposal
Training in Cardiopulmonary Resuscitation	X		X	X	X	X	X	X
Use of Virtual Reality	X	X	X	X	X	X	X	X
Development of a mobile application		X			X		X	X
Photorealistic interactive setting			X	X		X		X
Target audiences	Children	Paramedics	General	General	General	General	General	General
Development Platform	Oculus Rift	Oculus Quest	Oculus Rift	HTC Vive	GoogleCardboard	HTC Vive	Mobile	Google Cardboard
Sensor type	Leap Motion	Track Motion	Force plate	Load-cell	Depth an rate	Depth an rate	Depth an rate	Load-cell

**Table 2 sensors-21-02504-t002:** Values of the variables to be analyzed to identify.

Factor	Low	High
Previous Training	Not-Trained	Trained
Frequency (f)	100 compressions/min	150 compressions/min
Suggestions (sug)	Disabled	Active
Indicator (indic)	Disabled	Active

**Table 3 sensors-21-02504-t003:** Analysis of variance table for the percentage of correct compressions given for all the variables. The number in red highlights the best *p*-value of the experiment.

Source	Sum of Squares	Degrees of Freedom	Mean Square	F-Ratio	*p*-Value
Main Effects					
A: Previous Training	1444.53	1	1444.53	13.71	0.001
B: Frequency	7.03125	1	7.03125	0.07	0.7981
C: Suggestions	19.5313	1	19.5313	0.19	0.6703
D: Indicator	26.2813	1	26.2813	0.25	0.6216
Residuals	2845.59	27	105.392		
Total	4342.97	31			

**Table 4 sensors-21-02504-t004:** Analysis of variance for the percentage of correct compressions given for the previous training variable. The number in red highlights the best *p*-value of the experiment.

Source	Sum of Squares	Degrees of Freedom	Mean Square	F-Ratio	*p*-Value
Main Effects					
A: Previous Training	1444.53	1	1444.53	14.95	0.0006
Residuals	2898.44	30	96.6146		
Total	4342.97	31

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
