# Peer review of "VirtualCPR: Virtual Reality Mobile Application for Training in Cardiopulmonary Resuscitation Techniques"

_sensors, 2021, doi:10.3390/s21072504_

Round 1

Reviewer 1 Report

This paper presents a mobile VR system for the purpose of CPR training, also employing a force sensor to be integrated in training mannequins.

The motivation behind this work is clearly presented, as well as background information regarding CPR and VR. The system is then presented, as well as its experimental validation.

There are a few concerns with regard to the work described, which should be addressed in the revised manuscript.

First, there are several research efforts reported in literature for CPR training with VR, some of which employ mobile VR. Therefore, the related work section should be expanded to further elaborate on such related works.

More importantly, however, the specific contribution of the proposed work is not made clear. Is it that it does not need specialized equipment? Is it that it proposes an alternative training method? Currently, this is quite ambiguous to the reader.

Furthermore, the experiment that has been carried out should be targeted at supporting authors' claims with regard to the innovative aspects of the proposed system. Currently, the experiment mainly proves that "expert users concentrate the highest percentage of correct compressions", a finding which is pretty much expected and does not prove anything with regard to the training qualities of the proposed system. The authors are strongly suggested to carry out additional experiments with target users (four participants is an extremely small sample after all). These experiments should focus on answering a specific research question (what is it that the authors wish to measure / prove?) and should be designed accordingly. 

With regard to the experiment description, please note that User experience (UX) is a term used to denote the overall experience of the user with an interactive system. For the experiment carried out, participants' prior expertise with CPR was studied and not user experience.

Abstract starts by repeating "rescue breaths" a couple of times.

Reviewer 2 Report

The paper addresses an interesting concept,"VirtualCPR: Virtual reality mobile application for training in Cardiopulmonary Resuscitation Techniques". The authors have invested considerable thought and effort into the problem investigated. The work is almost well-understood and falls within the journal's scope, though in some parts, the writing deteriorates considerably and needs some amendments. Overall, the paper is interesting from the practical and theoretical points of view. There are, however, some deficiencies with the paper, which should be addressed/responded before it can be recommended for publication.

  • Most of your sentences are too long. Some major sentences do not have references. The article needs one last round of proofreading.
  • In the introduction section: Please try to refer to the latest available numbers. “Figure 1”
  • This sentence needs a proper reference “Hence, heart disease for 2018 accounted for more deaths in Mexico than cancer 34 and homicides combined.”
  • As VR is one of the significant parts of this research, the authors strongly suggested adding information regarding usages of VR in other industries to define VR suitably and then work on your goal, which can help readers better understand this technology. You can add this information in the Introduction section (Line 57-62) or section 2. Hoping to help in this task, I provide some literature:

- “Systematic Review on Which Analytics and Learning Methodologies Are Applied in Primary and Secondary Education in the Learning of Robotics Sensors”, Sensors, 2021 - mdpi.com

-       Development of a Virtual Reality Simulator for an Intelligent Robotic System Used in Ankle Rehabilitation, Sensors, 2021 - mdpi.com

 - “From BIM to extended reality in AEC industry”, Automation in Construction, 2020

-       The Effect of Virtual Reality Exercise Program on Sitting Balance Ability of Spinal Cord Injury Patients, Sensors, 2021 - mdpi.com

  • The structure of sections 2 and 3 are not clear. Please restructure these sections. 
  • Lines 252-258: There are various researches regarding VR in 2020 and 2021. These technologies are changing daily so you have to use the most recent articles.
  • Section 4 (Design of VirtualCPR): I suggest having a diagram that shows your complete methodology and then describes them step by step. 
  • The format, size, and colors need to be changed in all figures. Try to draw simple with fewer spaces around the boxes. 
  • It would be best to define why the authors preferred VR and not Augmented Reality (AR) for this procedure. What are the differences, advantages, or if AR can be better for your aim, you can suggest it for future research. 
  • You do not have any 2021 references. Please try to change your references to the latest ones.
  • All figures need to describe in the text adequately.
  • The paper is too long. Please try to summarize some parts in tables to illustrate your main goals in each section.
  • The authors strongly suggested adding a “practical implications” section before the Conclusion section.
  • The result and conclusion sections need to be improved. The advantages and challenges need to be discussed. 

Reviewer 3 Report

This work proposes VirtualCPR, which is a mobile virtual reality application to support learning and practicing of basic technique of Cardiopulmonary Resuscitation (CPR) for expert or inexperienced in CPR. I read the manuscript with great interest and believe its topic is important and relevant. The authors performed a careful and thorough review of the literature, as the section was very informative and substantial. Appropriate theoretical framework was applied. I found the methodological part to be well justified and reasonable for this type of analysis. Although the manuscript is overall well-written and structured, it might benefit from additional spell/language checking. However,

Comments

First four words of abstract needs clarification i.e. “rescue breaths rescue breaths”

The introduction is not clear and very less literature is used. Follow these instruction: The introduction should briefly place the study in a broad context and highlight why it is important. It should define the purpose of the work and its significance, including specific hypotheses being tested. The current state of the research field should be reviewed carefully and key publications cited. Please highlight controversial and diverging hypotheses when necessary. Finally, briefly mention the main aim of the work and highlight the main conclusions. Keep the introduction comprehensible to scientists working outside the topic of the paper.

Lines 110 to 120, authors just citied the previous authors. Explain what technique previous authors other used, objectives of the study, methodology, their results etc.

How authors get equation 1? Add some more explanation.

Authors should further clarify and elaborate novelty in their contribution.

What are the limitations of the present work?

Add one paragraph about implications of the research.

Below papers has some interesting implications that you could discuss in your introduction and how it relates to your work.

  • Ijaz, Muhammad Fazal, Muhammad Attique, and Youngdoo Son. "Data-Driven Cervical Cancer Prediction Model with Outlier Detection and Over-Sampling Methods." Sensors 20.10 (2020): 2809.
  • Ali, Farman, et al. "A smart healthcare monitoring system for heart disease prediction based on ensemble deep learning and feature fusion." Information Fusion 63 (2020): 208-222.

Round 2

Reviewer 1 Report

In the revised version the authors have added a table to summarize how they compare with other efforts in the field.

However, taking into account that this is a journal publication, the authors are suggested to expand their related work section, by including other works (and studies) as well. Indicative works that the authors could study are:
Leary, M., McGovern, S. K., Chaudhary, Z., Patel, J., Abella, B. S., & Blewer, A. L. (2019). Comparing bystander response to a sudden cardiac arrest using a virtual reality CPR training mobile app versus a standard CPR training mobile app. Resuscitation, 139, 167-173.

Everson, T., Joordens, M., Forbes, H., & Horan, B. (2021). Virtual Reality and Haptic Cardiopulmonary Resuscitation Training Approaches: A Review. IEEE Systems Journal.

Semeraro, F., Ristagno, G., Giulini, G., Gnudi, T., Kayal, J. S., Monesi, A., ... & Scapigliati, A. (2019). Virtual reality cardiopulmonary resuscitation (CPR): Comparison with a standard CPR training mannequin. Resuscitation, 135, 234-235.

Nas, J., Thannhauser, J., Vart, P., van Geuns, R. J., van Royen, N., Bonnes, J. L., & Brouwer, M. A. (2019). Rationale and design of the Lowlands Saves Lives trial: a randomised trial to compare CPR quality and long-term attitude towards CPR performance between face-to-face and virtual reality training with the Lifesaver VR app. BMJ open, 9(11), e033648.

It is also important to note that the claims regarding the findings of the experiment are not accurate. In particular, the authors claim that "Thus, the proposed virtual reality mobile application would help the learning and practice of the CPR technique in an interactive virtual setting for users with and without previous training, since the significance of this variable would increase when using VirtualCPR continuously".

In fact, what the experiment revealed is that individuals who were already experienced in CPR performed better. There was no study on whether experience in practicing CPR increases with the current application. Therefore, the aforementioned claim is unsupported and should be rephrased.

It would be more practical to try - if possible - to point out other factors or observations that came out of the experiment, e.g. with regard to users' interaction, system feedback, etc.

Finally, since additional experiments cannot be carried out (at least for the moment), the paper should acknowledge in the limitations section that the sample of participants is too small. 

Reviewer 2 Report

"Accept in present form"

Reviewer 3 Report

.
